



# Climate change projections of wet and dry extreme events in the Upper Jhelum Basin using a multivariate drought index: Evaluation of bias correction

Rubina Ansari[1], Ana Casanueva[2,3], Muhammad Usman Liaqat[1], Giovanna Grossi[1]

[1] Department of Civil, Environmental, Architectural Engineering and Mathematics, University of Brescia, Italy

[2]Dept. Matemática Aplicada y Ciencias de la Computación (MACC), Universidad de Cantabria, Santander (Spain)

[3]Grupo de Meteorología y Computación (Unidad Asociada al CSIC por el IFCA), Santander (Spain)

Corresponding author's Email: r.ansari@unibs.it

**Abstract**

Bias correction (BC) is often a necessity to improve the applicability of global and regional climate model (GCM and RCM, respectively) outputs to impact assessment studies, which usually depend on multiple potentially dependent variables. To date, various BC methods have been developed which adjust climate variables separately (univariate BC) or jointly (multivariate BC) prior to their application in impact studies (i.e., the component-wise approach). Another possible approach is to first calculate the multivariate hazard index from the original, biased simulations, and bias-correct the impact model output or index itself using univariate methods (direct approach). This has the advantage of circumventing the difficulties associated with correcting the inter-variable dependence of climate variables which is not considered by univariate BC methods.

Using a multivariate drought index (i.e., SPEI) as an example, the present study compares different state-of-the-art BC methods (univariate and multivariate) and BC approaches (direct and component-wise) applied to climate model simulations stemming from different experiments at different spatial resolutions (namely CORDEX, CORDEX-CORE and CMIP6). The BC methods are calibrated and evaluated over the same historical period (1986-2005). The proposed framework is demonstrated as a case study over a transboundary watershed, i.e. the Upper Jhelum Basin (UJB) in the Western Himalaya.

Results show that (1) there is some added value of multivariate BC methods over the univariate methods in adjusting the inter-variable relationship, however, comparable performance is found for SPEI indices. (2) The best performing BC methods exhibits a comparable performance under both approaches with a slightly better performance for the direct approach. (3) The added value of the high-resolution experiments (CORDEX-CORE) compared to their coarser resolution counterparts (CORDEX) are not apparent in this study.

**Key words:** Bias correction, CMIP6, CORDEX, CORDEX-CORE, SPEI indices, model evaluation

## 1. Introduction

Weather and climate related extreme events (floods, droughts, heatwaves, storms, etc.) that arise from complex interactions of various physical processes across multiple scales in space and time, are projected to be amplified under global warming conditions and thus are expected to create huge societal and ecological impacts (Kopp et al., 2017; Zscheischler et al., 2018; Raymond et al., 2020; Zscheischler et al., 2020). Such projected climate assessments are usually undertaken using impact models or hazard indices under different global warming scenarios. Several hazard indices and impact models have been developed according to the needs of different sectors and they are usually based on one or more essential climate variables (ECV). For instance, maximum consecutive five-day precipitation and the number of days with minimum temperature above 20 °C rely on one ECV only i.e., precipitation and temperature, respectively, while some require more complex calculations (e.g., the river flow index using runoff based on the results of hydrological model simulations).





Several studies employ such indices and impact models for the assessment of the sectorial impacts of climate change. For instance, drought indices (standardized precipitation index-SPI and standardized precipitation evapotranspiration index-SPEI) with implication in water related sectors, especially agriculture, hydrology and water management (Maru et al., 2021; Ansari and Grossi, 2022), snow indices (Snow days and Mean winter snow depth) in the context of water management, ecology, tourism, or road maintenance (Schmucki et al., 2017), river flow indices (100-year return level of daily high streamflow and 10-year return level of 7-day average low streamflow) for reservoir operation, energy production, flood and drought management (Naz et al., 2018), etc.

To assess the climate change impacts through such indices or any impact models, there is the need to have good quality observations and an adequate number of climate model simulations to characterize uncertainties at sufficiently high resolution to provide tailored regional to local climate information for impact assessment.

Global climate models (GCMs), that are the major source of knowledge about future climate change, represent a substantially simplified form of physical processes connecting the atmosphere, ocean, sea ice, land surface, and biogeochemical system. However, they typically present systematic biases with respect to observations (Christensen et al., 2008). These biases may be due to the temporal and spatial discretization (Teutschbein and Seibert, 2012), imperfect and unresolved representation of basic physical processes (Stevens and Bony, 2013) and parametrizations of unresolved sub-grid-scale processes (cloud formation, temperature inversion, convection, and precipitation, etc.). Even though regional climate models (RCMs) improve the representation of regional-scale processes up to some extent, their horizontal resolution is still coarser than the required for impact studies, and additionally suffer from substantial biases, partly inherited from the driving GCMs (Maraun et al., 2017). The use of raw GCM and RCM output for subsequent impact studies without any post processing could lead to ill-informed adaptation decisions for the foreseeable future. Nowadays global models from the 6th Coupled Intercomparison Project (CMIP6, (Eyring et al., 2015)) and regional counterparts from the Coordinated Regional Downscaling Experiment (CORDEX, (Giorgi et al., 2009; Jones, 2010)) constitute the state-of-the-art simulations for global and regional climate, respectively. Within CORDEX, standard simulations are developed on a 0.44°x0.44° grid (approximately 50x50 km) for many spatially distributed domains and, more recently, the Coordinated Output for Regional Evaluations (CORDEX-CORE, (Teichmann et al., 2021)) provides a reduced set of models on a higher resolution grid (approx. 25x25 km) covering most of the continental domains.

Bias correction (BC, also known as bias adjustment) is commonly applied to climate model output as a post-processing step to render climate model output more useful for climate impact studies. Over the recent years, a number of bias correction methods has been developed, varying from simple adjustments of the mean to correction of all quantiles, either univariate or multivariate, trend-preserving or not. These methods can only reduce systematic biases resulting from subgrid-scale parameterizations and unresolved orography under the current climate, but their efficiency is constrained by the misrepresentation of basic physical processes in the models, such as large-scale atmospheric circulation (Eden et al., 2012; Addor et al., 2016; Maraun et al., 2017). Further, BC constitutes an additional source of uncertainty in century-long climate change projections when applied under the stationarity (time invariance) assumption (Christensen et al., 2008; Ehret et al., 2012) and thus may induce physically implausible climate future signals (Maraun et al., 2017). Since BC might introduce inconsistencies in the bias-corrected data considerable attention should be paid towards its evaluation not only in terms of simulated statistical moments but also regarding trend preservation and inter-variable physical coherence. The latter is especially important for any climate index or impact model whose calculation depends on more than one variable (e.g., multivariate drought indices, fire weather indices, ecological and hydrological models, etc.). For example, the physical coherence between precipitation and temperature determines the available water for evaporation over arid and tropical watersheds and affects the snow accumulation and melting processes (Chen et al., 2018; Guo et al., 2020).

The physical coherence among several meteorological variables and their dynamic nature in the projected climate is confronted with growing discussion under the BC framework. Contrasting reviews are found in literature. Various researchers advocate the use of multivariate BC methods to reconstruct the inter-variable coherence of the



observations to the simulated climatic data (Zscheischler et al., 2019; François et al., 2020; Guo et al., 2020). For instance, François et al. (2020) report the added value of multivariate BC methods over univariate ones and conclude that the choice of the BC method should be based on the end user's goal. Conversely, Räty et al. (2018) find that the univariate and multivariate methods perform similarly, while Wilcke et al. (2013) show that univariate bias adjustment

is able to retain the quality of the temporal structure and the inter-variable dependencies of the uncorrected data. However, it is also argued that the ability of climate models to respond in a physically consistent way to external forcings is one of their basic foundations (Wilby et al., 2000) and that relationships between climate variables are not constant over time (time-invariant) (Mahony and Cannon, 2018; Hao et al., 2019). Another alternative approach in practice is the direct correction of the multivariate index (Casanueva et al., 2014; Casanueva et al., 2018; Li et al.,

2019; Chen et al., 2021). This direct approach allows the preservation of the physical and temporal coherence among the primary variables as represented in the original climate model output. However, it may hide compensating biases in the contributing variables, particularly in the case of complex indices bearing in their formulation non-linear relationships between components (Casanueva et al., 2018; Van De Velde et al., 2020).

In this work we intercompare different state-of-the-art BC methods (univariate and multivariate) and BC

approaches (direct and component twise) applied to climate model simulations stemming from three modeling initiatives (CMIP6, CORDEX -WAS-44 domain- and CORDEX-CORE -WAS-22 domain) for a multivariate drought index (namely the Standardized Precipitation Evapotranspiration Index, SPEI). The performance of BC and climate model simulations is examined in terms of inter-variable physical coherence of involved key essential variables i.e., precipitation (Pr), maximum temperature (Tmax) and minimum temperature (Tmin), and characteristics of extreme

events (duration, severity and frequency of wet and dry events) during the historical period 1986-2005. The proposed framework is demonstrated as a case study over a transboundary watershed, namely the Upper Jhelum Basin (UJB) located at the foothills of Western Himalaya.

The objectives of the study are:
– To assess the added value of multivariate bias correction methods with respect to the univariate bias correction

methods in the context of physical coherence of two variables, i.e., multivariate dependency.
– To assess the applicability of the direct and component-wise bias correction of a multivariate index (SPEI).
– To assess the added value of the CORDEX-CORE simulations compared to the CORDEX counterparts, as well as the added value of CORDEX compared to CMIP6 after bias correction.

## 1.  Data and Methods

### 1.1 Standardized Precipitation Evapotranspiration Index (SPEI)

A multivariate drought index i.e., standardized precipitation evapotranspiration index-SPEI (Vicente-Serrano et al., 2010), is widely used to monitor and assess drought and their sectorial impacts under global warming conditions. It can be interpreted as the number of standard deviations by which the observed anomaly deviates from the long-term

mean. Various researchers highlighted its suitability to detect the onset and spatio-temporal evolution of drought at the regional to global scales (Wang et al., 2014; Ansari and Grossi, 2022), and recommended it for operational drought monitoring (Vicente-Serrano et al., 2010).

In the present study, SPEI is calculated using a 30-day accumulation period at daily time step which can be used for short- or long-term extreme events analysis. The calculation and application of daily SPEI are similar to that

of monthly SPEI except for the temporal resolution of input climatic data. Its calculation requires two parameters i.e., precipitation and potential evapotranspiration (PET). The latter involves numerous variables, including air surface temperature, air humidity, shortwave incoming radiation, water vapor pressure, and ground–atmosphere latent and sensible heat fluxes (Allen et al., 1998), which hinders its correct estimation. Various methods (physical or empirical) have been developed to indirectly estimate PET from meteorological variables. These methods also vary in their input

data requirement. The data-intensive methods such as the Penman–Monteith method, in general, provide better results than others for PET quantification (Droogers and Allen, 2002). However, the purpose of including PET in the drought index calculation is to obtain a relative temporal estimation, and therefore the method used to calculate the PET is not





critical (Vicente-Serrano et al., 2010). A study conducted by Beguería et al. (2014) compared the SPEI values using three different methods for PET estimation (Penman- Monteith, Hargreaves, and Thornthwaite) and found small differences in humid regions. Mavromatis (2007) also found similar results for a drought index (i.e., Palmer Drought Severity Index-PDSI) when considering simple and complex PET methods. Therefore, the present study employs a

simple temperature-based Hargreaves-Samani method (involves Tmax and Tmin) (Hargreaves and Samani, 1985) due to data availability. The SPEI calculation involves two further steps: aggregation of daily climatic water balance time series at different time scales (30 days in the present study), and then its normalization into a log–logistic probability distribution to obtain the SPEI index series. The log-logistic distribution for SPEI calculation is used and recommended by many researchers (Vicente-Serrano et al., 2010; Wang et al., 2015; Himayoun and Roshni, 2019;

Ansari and Grossi, 2022). A more detailed description of the SPEI calculation procedure can be found in (Vicente-Serrano et al., 2010) and (Wang et al., 2015).

### 2.2 Identification of extreme events and their characteristics

The wet and dry extreme events are identified by using monthly SPEI values (computed from daily SPEI values, see Sect. 2.1). Although the SPEI was originally proposed for drought monitoring, it can also be used as a tool

to detect flood risk, since it quantifies both positive and negative anomalies representing wet and dry conditions, respectively. In the present study, wet and dry extreme events are defined as the positive (SPEI ≥ 1) and negative SPEI (SPEI ≤ −1) for at least two consecutive months, respectively. Three event indices (severity, duration, and frequency) are considered to characterize the wet and dry extreme events during the historical period (1986-2005). The duration of a wet/dry event (denoted hereafter as wet duration -WD- and dry duration-DD) is the number of consecutive months

with SPEI above/below 1; severity (wet severity -WS- and dry severity-DS) refers to the cumulative value of the index from the first month to the last month of the wet/dry event and it represents the water surplus and deficit, respectively; and the absolute frequency (wet frequency -WF- and dry frequency-DF) is the total number of events in a specified time period. Since duration and severity are obtained for individual events, we consider the median value across all the identified events as the single index.

### 2.3 Reference Dataset

Because of complex orography, severe weather, and harsh environmental conditions in the High Mountains of Asia (HMA), observations from meteorological stations are rare in this region. Available weather stations are usually sparse and unevenly distributed. Gridded data, satellite observations and reanalysis are mostly used as an alternative, even though they are affected by the uncertainties inherent to the observations and to statistical post

processing (e.g., interpolation). In the present study, the W5E5 dataset (Lange, 2019) is used as the observational reference for the training of the bias correction methods during the historical period (1986-2005). This dataset was developed under the Phase 3b of the Inter-Sectoral Impact Model Intercomparison Project (ISIMIP3b) and was used as reference to bias-correct the climate models output which serve as input data to carry out the impact assessments under ISIMIP3b. The W5E5 is a merged dataset, developed using version 2.0 of WFDE5 data (WATCH forcing data

methodology applied to ERA5 data (Weedon et al., 2014; Cucchi et al., 2020) over land and ERA5 (Hersbach et al., 2020) over the ocean. W5E5 is a global daily dataset available at 0.5° horizontal resolution covering the period 1979–2016. The W5E5 dataset provides twelve meteorological variables however, the present study employed three variables i.e., precipitation, daily maximum near surface air temperature and daily minimum near surface air temperature.

The use of W5E5 (WFDE5 data over land and ERA5 over ocean) for the present study is motivated by the numerous previous studies. For instance, the suitability of ERA5 and its slightly overestimation of precipitation over the study region (UJB), especially over the mountainous part of the basin, have been evaluated and acknowledged by several researchers (Ansari et al., 2022; (Baudouin et al., 2020; Arshad et al., 2021; Liaqat et al., 2022). These studies recommend performing the bias correction of ERA5 with localized data before its application in impact studies.

Moreover, the WFDEI dataset, which is the predecessor of WFDE5 but based on ERA-Interim reanalysis, has also been applied to the UJB and surrounding regions to alleviate the data scarcity issue across the transnational border (Lutz et al., 2016; Dahri et al., 2016; Azmat et al., 2018). The WFDE5 benefits from the higher spatial and temporal resolution and better representation of spatial variability of ERA5, compared to WFDEI which was generated by





interpolating the lower-resolution ERA-Interim reanalysis. An evaluation of both products against meteorological observations shows that, on average, WFDE5 has lower mean absolute error and higher correlation than WFDEI for all variables (Cucchi et al., 2020). W5E5 has also been used as the reference observational dataset for bias correction in the IPCC Interactive Atlas in the 6[th] Assessment Report (Gutiérrez et al. 2021).

## 2.4 Climate model simulations

In the present work, we consider climate model historical simulations from 2 GCMs of the 6[th] Coupled Intercomparison Project (CMIP6, (Eyring et al., 2015)), 17 RCM simulations (3 RCMs unevenly driven by 10 GCMs) of the coordinated Regional Climate Experiment (CORDEX, (Giorgi et al., 2009; Jones, 2010)) and 9 RCM

simulations (3 RCMs unevenly driven by 6 GCMs) of the CORDEX-COmmon Regional Experiment (CORDEX-CORE, (Teichmann et al., 2021)). For the RCM simulations, the south Asian domain (denoted as WAS) is considered. In particular, we use all available simulations by November 2021 for WAS-44 and WAS-22 domains (simulations conducted at horizontal resolutions of 0.44° and 0.22° on rotated grids, approximately 50km and 25km) for CORDEX and for CORDEX-CORE, respectively. The selection of this particular subset of GCMs from CMIP6 is motivated by

the availability of models with high spatial resolution (Table 1), approximately similar to the CORDEX counterparts. Coarser GCMs are not considered due to the small size of the catchment under study. Note that the driving GCMs of both CORDEX and CORDEX-CORE experiments are CMIP5 GCMs, thus emission scenarios vary across the different datasets, namely three shared socioeconomic pathways (SSPs) for the CMIP6 GCMs (SSP1-2.6, SSP2-4.5 and SSP5-8.5) and all available representative concentration pathways (RCPs) for CORDEX and CORDEX-CORE.

In particular, 8 RCMs provide climate projections under RCP2.6 and 9 RCMs under RCP8.5 within CORDEX-CORE, and 6 RCMs provide climate projections under RCP2.6 and 17 RCMs under RCP4.5 and RCP8.5 within the CORDEX experiment. Although some differences are found between SSPs and RCPs, the selected pairs approximately target a similar level of aggregated radiative forcing (Tebaldi et al., 2021). More details on the considered climate models and scenarios are given in Table 1 and 2.

**Table 1:** Details of the CMIP6 GCMs used in the present study

| Model | Institute (country) | Horizontal resolution | SSP126 | SSP245 | SSP585 |
|---|---|---|---|---|---|
| CNRM-CM6-1-HR | CNRM-CERFACS (France)<br><br>Centre National de Recherches Meteorologiques / Centre Europeen de Recherche et Formation Avancees en Calcul Scientifique | 0.5° × 0.5° | | × | × |
| EC-Earth3 | EC-Earth-Consortium | 0.6958702 × 0.703125 | × | × | × |

**Table 2:** Details of the CORDEX and CORDEX-CORE RCMs used in the present study

| Experiment | RCM | RCM description | Contributing CORDEX modelling center | Driving CMIP5 GCM | RCP26 | RCP45 | RCP85 |
|---|---|---|---|---|---|---|---|
| CORDEX (WAS-44) | RegCM4-4 | The Abdus Salam International Centre for Theoretical Physics (ICTP) Regional Climate Model version 4 (RegCM4; (Giorgi et al., 2012) | Centre for Climate Change Research (CCCR), Indian Institute of Tropical Meteorology (IITM), India | CCCma-CanESM2 | | × | × |
| | | | | CNRM-CERFACS | | × | × |
| | | | | CSIRO-QCCCE-CSIRO | | × | × |
| | | | | IPSL-IPSL-CM5A-LR | | × | × |
| | | | | MPI-M-MPI-ESM | | × | × |





| | | | | NOAA-GFDL-GFDL-ESM2M | | × | × |
|---|---|---|---|---|---|---|---|
| | RCA4 | Rossby Centre Regional Atmospheric Model version 4 (RCA4; (Samuelsson et al., 2011) | Rossby Centre, Swedish Meteorological and Hydrological Institute (SMHI), Sweden | CCCma-CanESM2 | | × | × |
| | | | | CNRM-CERFACS | | × | × |
| | | | | CSIRO-QCCCE-CSIRO | | × | × |
| | | | | ICHEC-EC-EARTH | × | × | × |
| | | | | IPSL-IPSL-CM5A-MR | | × | × |
| | | | | MIROC-MIROC5 | × | × | × |
| | | | | MOHC-HadGEM2-ES | × | × | × |
| | | | | MPI-M-MPI-ESM | × | × | × |
| | | | | NCC-NorESM1-M | × | × | × |
| | | | | NOAA-GFDL-GFDL-ESM2M | | × | × |
| | REMO2009 | MPI Regional model 2009 (REMO2009; (Teichmann et al., 2013) | Climate Service Center (CSC), Germany | MPI-M-MPI-ESM | × | × | × |
| CORDEX-CORE (WAS-22) | COSMO-crCLIM-v1-1 | (COnsortium for Small scale MOdelling) model (Baldauf et al., 2011) | Climate Limited-area Modelling (CLM) Community | ICHEC-EC-EARTH | | | × |
| | | | | MPI-M-MPI-ESM-LR | × | | × |
| | | | | NCC-NorESM1-M | × | | × |
| | RegCM4-7 | The Abdus Salam International Centre for Theoretical Physics (ICTP) Regional Climate Model version 4 (RegCM4; (Giorgi et al., 2012) | Centre for Climate Change Research (CCCR), Indian Institute of Tropical Meteorology (IITM), India | MIROC-MIROC5 | × | | × |
| | | | | NCC-NorESM1-M | × | | × |
| | | | | MPI-ESM-MR | × | | × |
| | REMO2015 | Climate Service Center Germany (GERICS). | Climate Service Center Germany (GERICS). | MOHC-HadGEM2-ES | × | | × |
| | | | | MPI-M-MPI-ESM-LR | × | | × |
| | | | | NCC-NorESM1-M | × | | × |

## 2.5 Bias Correction methods

Several univariate and multivariate bias correction methods are used in this study. A comparison between five univariate bias correction methods and three multivariate bias correction methods is performed with respect to their ability to reproduce observed univariate distributions and inter-variable relationships. The univariate methods





are applied to climate model simulations following two approaches: (1) individually to all involved essential climatic variables (i.e., the component-wise approach) and (2) directly to the uncorrected SPEI (i.e., the direct approach). All BC methods make a common assumption of stationary biases by applying the same calibrated transfer function in the calibration period (1986-2005) to the future projected climate which may lead to modifications of the raw model 5 climate change signals for non-trend preserving methods. The table 3 summarizes the considered BC approaches and methods.

**Table 3:** Bias correction approaches and methods employed in the present study

| Approach | Method | Name | Reference |
|---|---|---|---|
| Component-wise | Univariate | Empirical quantile mapping (**EQM**) | (Déqué, 2007) |
| | | Parametric quantile mapping (**PQM**) | (Piani et al., 2010) |
| | | Generalized Pareto parametric quantile mapping (**GPQM**) | (Vrac and Naveau, 2007) |
| | | Quantile Delta Mapping (**QDM**) | (Cannon et al., 2015) |
| | | Detrended quantile mapping (**DQM**) | (Cannon et al., 2015) |
| | Multivariate | Multivariate Bias Correction: Pearson version (**MBCp**) | (Cannon, 2016) |
| | | Multivariate Bias Correction: Spearman version (**MBCp**) | (Cannon, 2016) |
| | | Multivariate Bias Correction with N-dimensional probability density function transform (**MBCn**) | (Cannon, 2018) |
| Direct | Univariate | Empirical quantile mapping (**EQM**) | (Déqué, 2007) |
| | | Parametric quantile mapping (**PQM**) | (Piani et al., 2010) |
| | | Generalized Pareto parametric quantile mapping (**GPQM**) | (Vrac and Naveau, 2007) |
| | | Quantile Delta Mapping (**QDM**) | (Cannon et al., 2015) |
| | | Detrended quantile mapping (**DQM**) | (Cannon et al., 2015) |

## 2.5.1 Univariate Bias Correction Methods

10 Five univariate methods (either parametric or empirical) are considered in this study. The present study uses the implementation included in the R package downscaleR (Bedia et al., 2020) which is part of the R bundle climate4R (Iturbide et al., 2019).

**Empirical quantile mapping (EQM):** This method calibrates an empirical transfer function that matches 15 all quantiles of model empirical cumulative distribution function (CDF) to those of reference dataset. The values lying outside the calibration range are adjusted through constant extrapolation (first and last percentile corrections for values below and above the calibration range, respectively) (Themeßl et al., 2012). The method also adjusts the overestimation of wet or dry days frequency (defined as days with precipitation above or below 1mm in the reference dataset) in the model using, respectively, adjusted wet-day threshold and frequency adaptation proposed by (Themeßl 20 et al., 2012; Wilcke et al., 2013). If a model produces too many wet days, then the wet-day frequency is corrected in such a way that it matches the observed wet-day frequency. In case of overestimation of dry days in the model, then the frequency adaptation is made through the random sampling of the observed Gamma distribution into the simulated first bin (0–1 mm) in order to generate wet days.

**Parametric quantile mapping (PQM):** This method adjusts the theoretical CDF of the model output onto 25 the corresponding observed distribution via a parametric transfer function calibrated over the training period (Piani et al., 2010). Assumptions are made about the distribution of a particular variable (i.e., precipitation and temperature follow the Gamma and Gaussian distribution, respectively). As for EQM, in the considered implementation the overestimation of wet or dry days in the model data is also adjusted using wet-day frequency correction and frequency adaptation approach, respectively.



**Generalized Pareto parametric quantile mapping (GPQM):** The method is specifically designed to adjust the extremes of the distribution. It fits two different parametric distributions to adjust the extreme and non-extreme values separately. The Gamma or Gaussian distribution (for precipitation and temperature, respectively) adjust the central part whereas Generalized Pareto distributions are applied above the 95[th] and below 5[th] percentiles (Vrac and
Naveau, 2007). As for EQM, the wet-day frequency correction and frequency adaptation are applied.

**The Quantile Delta Mapping (QDM):** The method was first developed by (Li et al., 2010) and (Wang and Chen, 2014) as 'equidistant' and 'equiratio' quantile matching, respectively. The main idea is to preserve the trends of all quantiles of the simulated distribution. Later,(Cannon et al., 2015) termed both methods as QDM due to its similarity to a quantile delta change method. Firstly, model projections are detrended by quantile and quantile mapping
is applied to adjust systematic distributional biases relative to the observations. Then the removed projected trends are reintroduced to the bias-corrected quantiles. Thus, it ensures that the sensitivity of the underlying climate model remains unaffected by the bias correction (at least so far as quantiles are concerned).

**Detrended quantile mapping (DQM):** The method is similar to QDM, except that absolute or relative changes in the simulated mean are accounted for, rather than all modeled quantiles (Cannon et al., 2015). Hence, the
long-term mean (linear) trend is removed, and bias correction is applied to the detrended series by empirical quantile mapping using all quantiles to adjust systematic distributional biases relative to observation. Then the mean trend is reintroduced to the bias-corrected series. As DQM only preserves long-term mean trends, it does not ensure to preserve the simulated model trends at the tails of the distribution that define climate extremes (Casanueva et al., 2020).

## 2.5.2 Multivariate Bias Correction Methods

The three MBC methods used to adjust the inter-variable structure in this study are **MBCp, MBCr** (Cannon, 2016) and **MBCn** (Cannon, 2018). The MBCp and MBCr methods are the combination of two approaches: firstly, quantile delta mapping is applied to each variable individually, in order to correct the marginal distribution of the variables including the preservation of absolute (in case of temperature) or relative (in case of non-gaussian variables like precipitation) raw climate change signal, and, secondly, multivariate linear rescaling is applied (Bürger et al.,
2011), in order to adjust the dependence structure through an iterative application of the Cholesky decomposition of the covariance matrix. The Pearson correlation and Spearman rank correlation are used as covariance matrix in the MCBp and MCBr methods, respectively. These two steps are repeated until both the marginal distributions and specified correlation matrix converge to those of the reference dataset.

The MBCn algorithm, which is based on the N-dimensional probability density function transform, is adopted
from an image processing algorithm used to transfer color information(Pitie et al., 2005; Pitié et al., 2007). Unlike MCBp and MCBr methods, MBCn permits to transfer all statistical characteristics of the observed multivariate distribution to those of the climate model outputs. In MBCn, random orthogonal rotation matrices are applied to the observed and climate model data to partially decorrelate the climate variables before the QDM. It is then rotated back with the inverse random matrices. The process of rotation, QDM and back rotation are repeated iteratively until the
multivariate distribution of the historical climate model data converges to that of the reference data. The present study uses the implementation included in the R package "MBC" (https://cran.r-project.org/web/packages/MBC).

## 2.6 Experimental framework

In this study, the BC methods presented above are applied to adjust daily maximum temperature, minimum
temperature and precipitation of 28 (global and regional) climate model simulations (Tables 1 and 2) towards the W5E5 reference dataset. All BC methods are calibrated in the period 1986–2005, being the correction functions calculated separately for each month in order to account for biases varying along the year. These corrections are then applied to the same period in order to evaluate their performance in present climate. Although the calibration and evaluation periods are the same, our approach can be considered independent since the evaluated aspect (i.e., SPEI
indices) is not directly adjusted by the BC methods. All analyses are carried out at the spatial resolution of the W5E5 grid (regular 50×50 km). For this reason, all model simulations are remapped into the W5E5 grid using nearest neighbor interpolation. As a consequence, there will be aspects of the added value of the higher-resolution WAS-22 experiments (related to better-resolved, fine-scale processes) that can be smoothed out, but they may still be present





after remapping them onto a coarse resolution. For all BC methods, daily maximum temperatures, minimum temperature, and precipitation from each GCM/RCM are corrected independently at each grid box. Due to the multivariate nature of the SPEI, daily maximum temperature, minimum temperature and precipitation are corrected separately (in case of univariate BC methods) and jointly (in case of multivariate BC methods) prior to the SPEI

calculation (i.e., the component-wise approach; (Casanueva et al., 2018). An alternative to this approach is to first calculate the SPEI index from the original, biased simulations (i.e. original modeled intervariable relationships remain) and, secondly, bias-correct the index itself using univariate methods (direct approach; (Casanueva et al., 2018)). Note that normal distribution is assumed for the direct correction of SPEI through the PQM method. In addition to the evaluation of BC methods and approaches performance, the added value of higher spatial resolution in the modeled

data (CORDEX-CORE over CORDEX and CMIP6) is assessed. Both assessments are performed in terms of the ability to simulate the mean spatio-temporal distribution of SPEI and its derived indices over the study region.

## 2.7 Evaluation metrics

The performance of the raw and bias-corrected climate model simulations (component-wise approach) is firstly evaluated in terms of inter-variable relationships by using two statistical metrics, namely the correlation

coefficient (Pearson, 1895; Wilks, 2011) and Perkins skill score (Perkins et al., 2007). The correlation coefficient between daily time series of two variables (Spearman for Pr vs Tmax and Pr vs Tmin and Pearson for Tmax vs Tmin) is computed at each grid cell to measure the relationship between pairs of variables. Since Pearson and Spearman correlation coefficients imply a linear and non-linear relationship, respectively, are hence recommended for temperature and precipitation respectively (Wilcke et al., 2013). The Perkins skill score is a quantitative measure of

the similarity between two probability density functions (PDFs) by measuring the common area between them. A value of 0 indicates no overlap and a value of 1 indicates distributions are identical. In the present study, an extended version of the Perkins skill score with two dimensions is used, that accounts for the similarity (overlap) between the modeled joint distribution of two meteorological variables and the observed counterpart (Casanueva et al., 2019). Further, the raw and bias-corrected climate model data (component-wise and direct approaches) are evaluated by using

the mean bias (ratio of model to reference) of the SPEI indices (median duration, median severity, and absolute frequency; see Sect. 2.2).

## 3. Results

### 3.1 Evaluation of inter-variable relationships

To evaluate the inter-variable structures, correlation coefficient (COR) and Perkins skill score (PSS) between maximum and minimum temperatures are computed at each grid cell to measure the relationship between the two physical variables (Fig. 1). The heat map shows the spatially averaged values of Pearson correlation (size of the marker) and Perkins skill score (colored scale) for the raw and bias-corrected model output, i.e., the larger the marker the stronger the relationship between the two variables, and the yellower the color the more similar the joint PDFs are

to the reference dataset. The Spearman correlation coefficient between the other pairs of variables (Pr vs Tmax and Pr vs Tmin) for the reference dataset is found to be negligible (Fig. S1). Therefore, the ability of the BC methods to adjust the inter-variable dependencies is evaluated only for maximum and minimum temperature.

Overall, small differences in terms of correlation are found for raw and bias-corrected model output, compared to the reference value. Maximum and minimum temperature showed strong positive correlation exceeding

0.9 in W5E5, which is also evident in the raw models and preserved after BC. However, one climate simulation (REMO2009 RCM driven by MPI GCM) shows weaker correlation for the raw model output and improves with all BC methods.

Regarding PSS, low values for the raw model outputs show the differences in the joint PDF of Tmax and Tmin compared to the reference data, meaning that the inter-variable dependencies in the reference dataset are not

well presented by the raw model output. However, this inter-variable physical coherence improves up to some extent with the application of all BC methods. Among univariate BC methods, the empirical ones (EQM, DQM and QDM) performed better than the parametric counterparts in terms of the inter-variable physical coherence between maximum and minimum temperature. As expected, all MBC methods performed well and improve upon the univariate ones.



Among three MBC methods, MBCn outperformed the other two methods, with a very similar joint PDF to the reference data.

All the above holds for most of the different climate model simulations, regardless of the RCM, driving GCM, original spatial resolution (CORDEX vs CORDEX-CORE) and modelling experiment (CMIP vs CORDEX). Although the differences among different climate model simulations from three modelling experiments exist for parametric methods (PQM and GPQM), no specific pattern is found. Thus, no evident added value of the higher resolution experiment models (WAS-22) is observed over low-resolution experiment models (WAS-44 and CMIP6), either in correlation or Perkins skill score. However, a clear added value of multivariate BC methods is apparent.

To further explore the ability of raw and BC datasets to reproduce the reference full joint probability distribution of maximum temperature and minimum temperature, two-dimensional kernel density plots together with marginal histograms (Fig. 2) are developed for a single grid box of a RCM (highlighted red box in Fig. 1 and Fig.2). The selection of this particular grid box is motivated by the low values of correlation and PSS for the raw simulation and subsequent improvement after BC, in order to investigate whether low PSS values can be attributed to biases in maximum temperature or minimum temperature or both. Higher density values in the reference dataset (Fig. 2, first row and second column) take place around 20ºC and 7ºC for maximum and minimum temperature, respectively. For the raw model output, the shape and location of the joint distribution and maximum probability are biased at both ends of the distributions, as evident in low PSS value i.e., 0.641. This low PSS value of the raw simulation is attributed to both variables, but especially to the misrepresentation of the minimum temperature distribution. Likewise, the temporal correlation between the two variables is slightly lower than in the reference. Correlation and PSS improve after BC regardless the BC method with least improvement in the joint distribution with GPQM (PSS=0.707). Higher density values are well located with parametric methods i.e., PQM and GPQM (this would be expected, because they fit the mean and standard deviation in the calibration phase), but for GPQM they are no so differentiated as in the reference data. Other three univariate BC methods (i.e., EQM, DQM and QDM) and multivariate methods improve the representation of the joint distribution in a similar way, although maxima are no so differentiated as in the reference data.

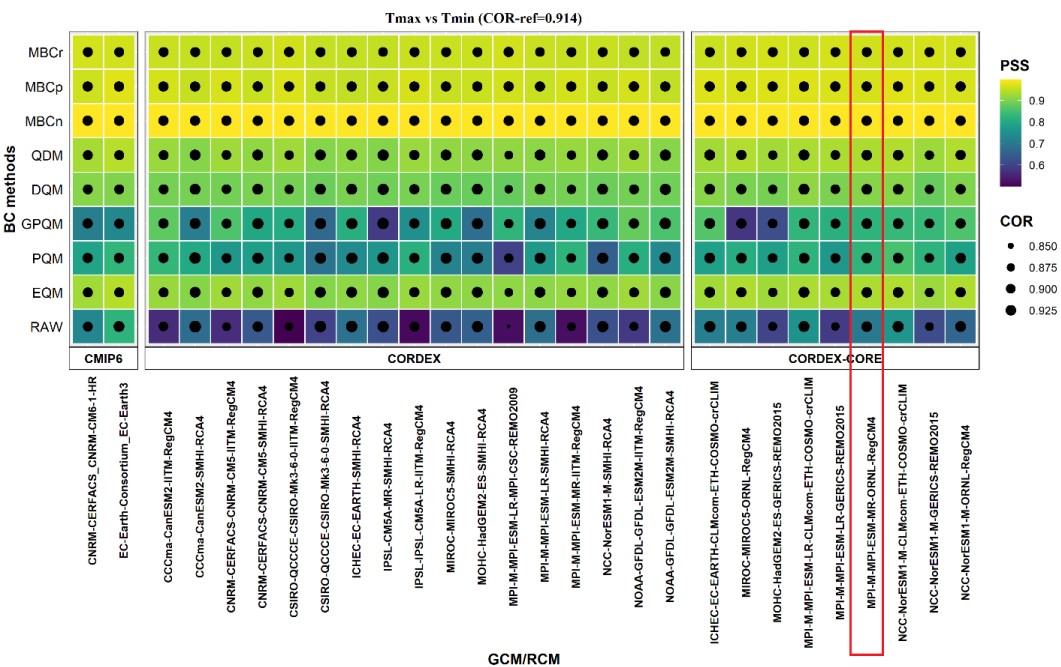





**Figure 1:** Pearson correlation coefficient (COR, circle) between maximum and minimum temperatures and Perkins skill score (PSS, colored scale) of the joint PDFs for the raw and bias-corrected climate model data (only component-wise approach). Correlation for the reference dataset is shown on top of the panel. The highlighted RCM (red square) is selected for further analyses in Fig. 2.



**Figure 2:** Two-dimensional kernel density plots for the highlighted grid box (red box). Blue histograms (and X-axis) refer to minimum temperature and red histograms (and Y-axis) refer to maximum temperature. Shadings represent the 2-D density distribution for the reference, raw and eight BC methods. Contour lines represent the probabilities in the reference dataset, which are overlaid on the model probabilities for the sake of comparison. COR depicts the Pearson correlation coefficient between daily minimum and maximum temperatures and PSS represents the two-dimensional Perkins skill score of distributional similarity.



### 3.2 Evaluation of SPEI characteristics

The performance of the raw and bias-corrected climate model simulations is evaluated in terms of mean biases (ratio of model to reference dataset) in SPEI indices (duration, severity, and frequency, see Sect. 2.2) during the historical period (1986-2005). The spatial distribution of biases calculated from multi-model ensemble mean SPEI indices, separately for CMIP6 (2 simulations), CORDEX (17) and CORDEX-CORE (9) are presented in Figs. 3-4 and S2-S5. Results show that the northeast part of the region, located at the foothills of Western Himalayas, is found to be more affected by wet and dry events with higher severity and duration (see the upper left panel in each Figure). The higher susceptibility of the region towards more extreme events could be explained with the increasing rates of global warming over mountainous region i.e., Western Himalayas, also reported by many researchers (Pachauri et al., 2014; Zaz et al., 2019; Rashid et al., 2020; Shafiq et al., 2020; Ansari and Grossi, 2022). Studies by (Negi et al., 2018) and (Dimri and Dash, 2012) also confirm that most of the western Himalayan region recorded a significant warming trend especially from 1975 onwards. This is also supported by the tree-ring chronologies of the region which indicate rapid growth of the tree rings in the recent decades especially at higher altitudes (Borgaonkar et al., 2009).

In the context of biases, different sign biases are found depending on the location and SPEI index. The underestimation of all SPEI indices is higher in the northeast part of the region which shows that climate models performance is relatively poor over mountainous regions. Overall, larger biases are found in frequency indices as compared to duration and severity indices. These under- and overestimations are partly alleviated by most of the bias correction methods. Regarding severity indices (Fig. 3 and 4), remaining biases after BC are similar across BC methods, with underestimation of WS in the mountainous region and no specific pattern for DS. In case of duration indices (Fig. S2 and S3), the overestimation at low lands is improved by all BC methods, however that improvement is not only negligible but also degrade the raw CMIP6 ensemble in the northeast of the basin (mountainous region). The underestimation of frequency indices over high mountains (Fig. S4 and S5) is partially elevated by all BC methods with slightly better performance under the direct approach. However, the bias correction induced an overestimation in WF over low lands which is not present in raw ensembles of all datasets. Overall, GPQM is found to bring the least improvement for most of the SPEI indices and the added value of MBC methods is not evident over remaining univariate methods.

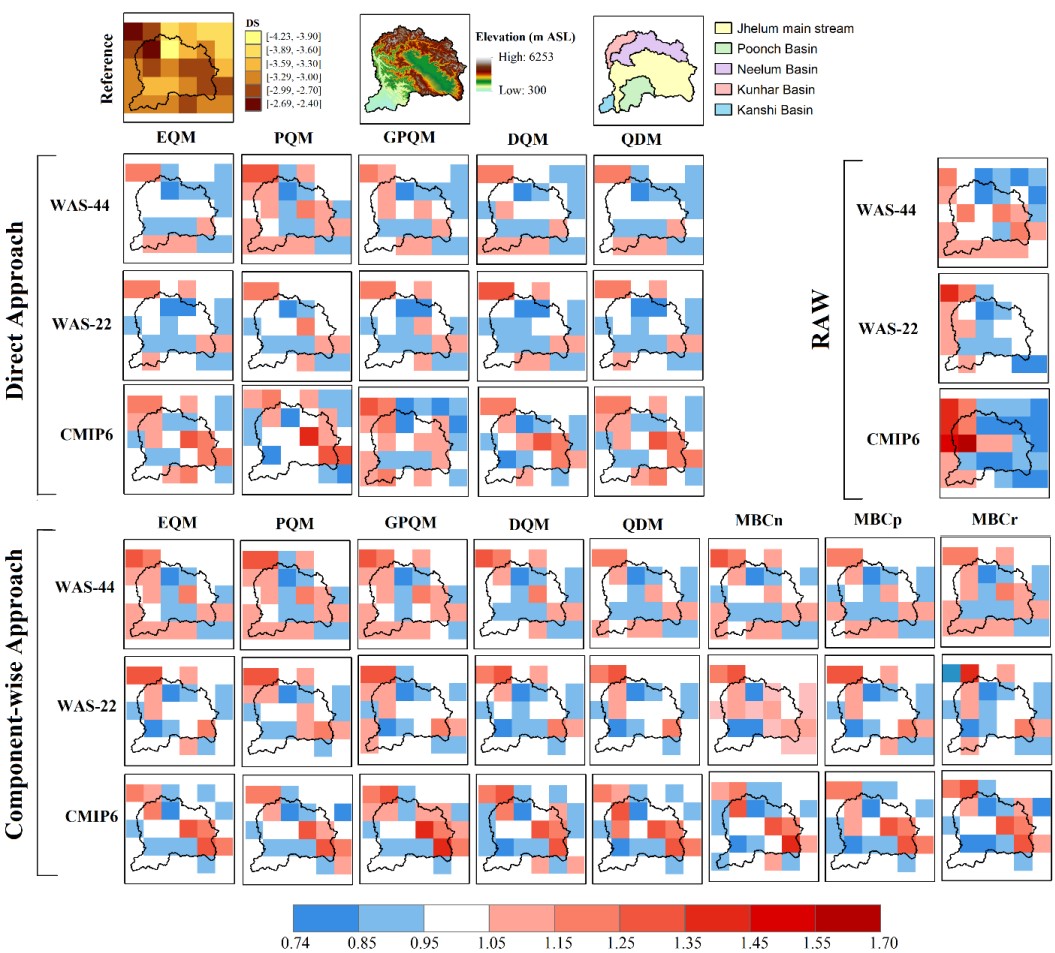

**Figure 3:** Median dry severity (DS) in the reference dataset expressed in accumulated SPEI units (first row, left), digital elevation model in meter above sea level (first row, center and location of sub-basins (first row, right), and biases (as a ratio of model to reference) of DS for the multi-model raw and bias-corrected ensembles, for the two bias correction approaches and all methods (columns).



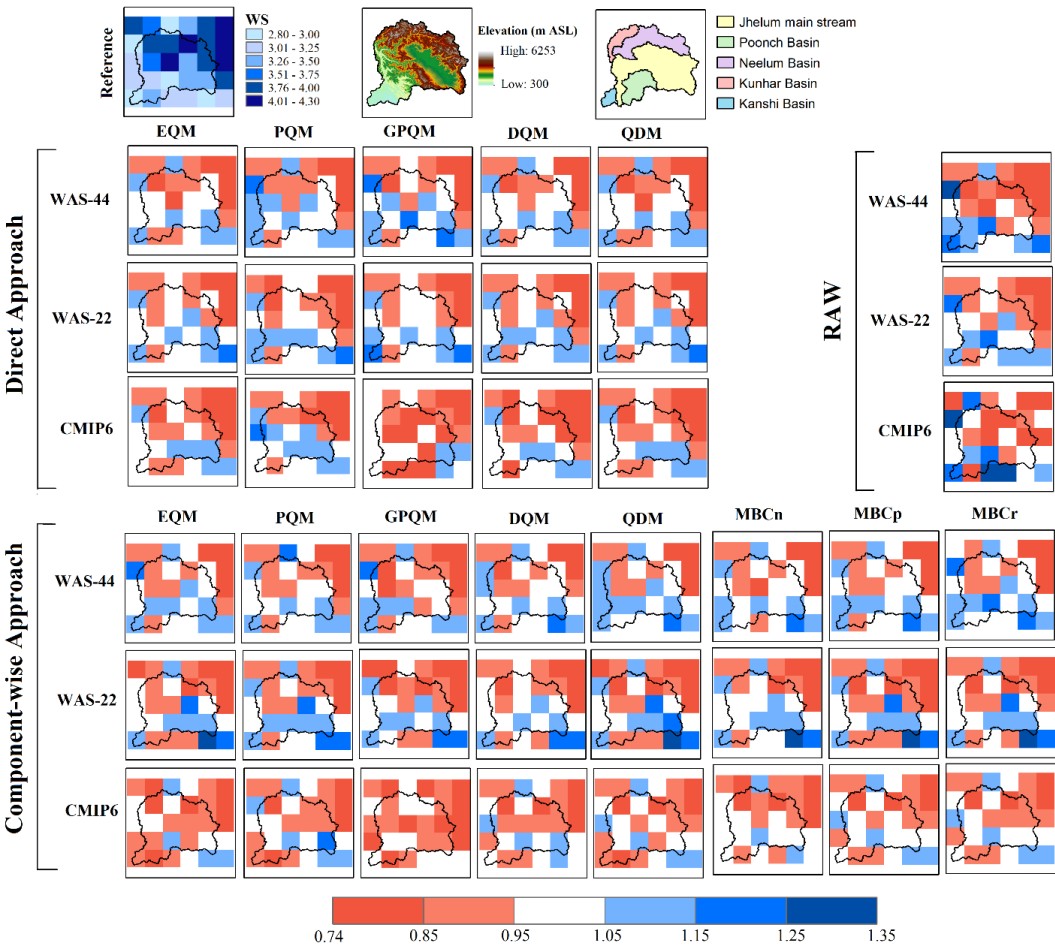

**Figure 4:** Same as figure 3, but for median wet severity (WS), expressed in accumulated SPEI units.

The regionally averaged biases in median duration and severity, and absolute frequency of dry and wet events
for all individual climate simulations (CORDEX-CORE, CORDEX and CMIP6) are summarized in Fig.5. Overall,
results show that raw models generally underestimate all SPEI indices and bias correction alleviates this for frequency
and severity indices, but still shorter events than in the reference dataset are found after the corrections. The
conclusions drawn from the spatial plots also relate to climate models spread. For instance, the small improvement in
duration indices over high mountains is in line with a slight reduction of the model spread after bias correction, yet
the underestimation of the DD and WD remains. Similarly, for WF the reduction in the underestimation over high
mountains and induction of overestimation over low lands by all BC methods is in agreement with changes in the
models spread after bias correction (Fig. S5 and 5).
   In general, all BC methods under the direct approach present similar improvements for all SPEI indices,
except PQM for dry extremes which shows smaller improvements for some simulations. Regarding the component-
wise approach, the empirical BC methods i.e., EQM, DQM and QDM performed relatively better than PQM and
GPQM for most of the models and SPEI indices. Small differences are found in the performance of three MBC
methods.





Regarding the spatial resolution, no obvious benefit of the higher resolution (CORDEX-CORE vs. CORDEX and CMIP6) is apparent. Raw model outputs from CORDEX-WAS44 show more spread than the CMIP6 and CORDEX-CORE experiment models, which could be due to the number of simulations. After BC, the spread of CORDEX and CORDEX-CORE is similar and thus, no clear added value of higher resolution is found.

5    Overall, the performance of BC methods and climate models are found to be relatively better for drought indices than for flood indices. Most of the models underestimate wet duration and severity over the region before and after bias correction.

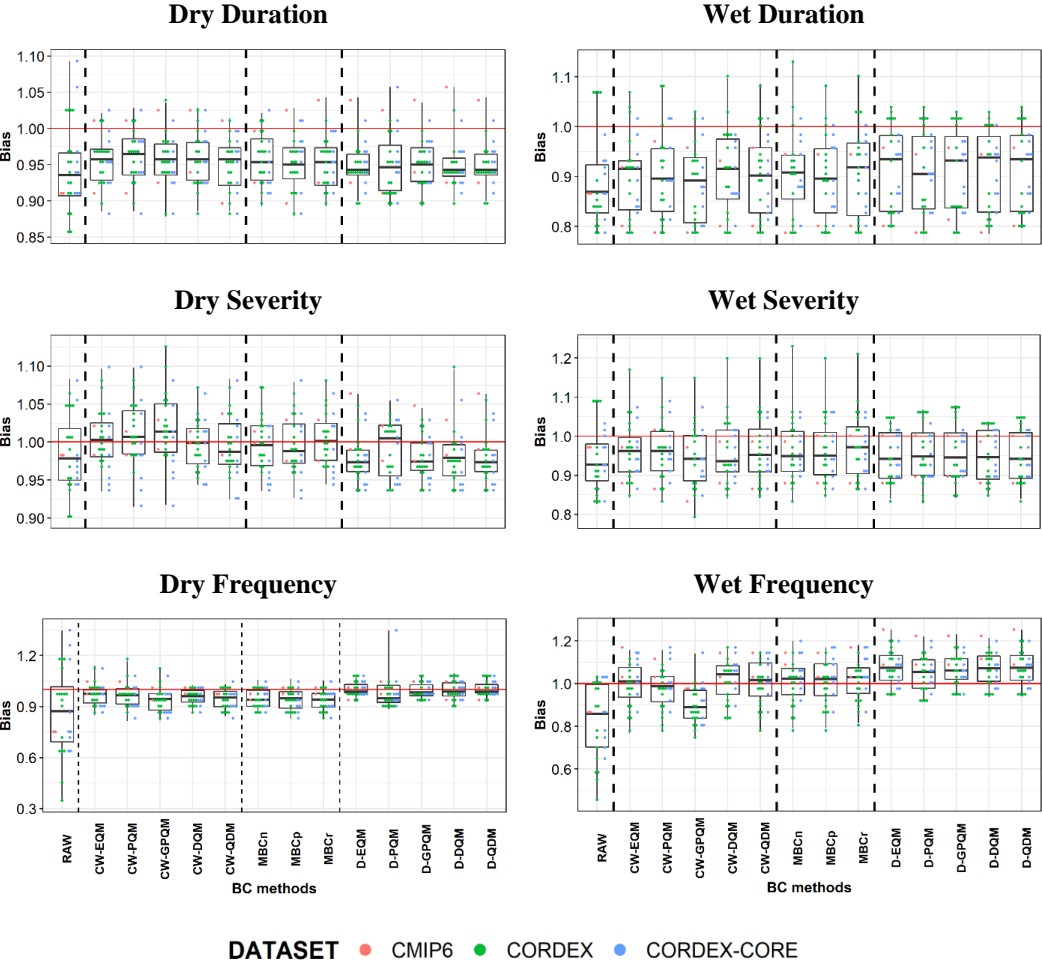

10    **Figure 5:** Biases (as a ratio of model to reference) in spatially averaged SPEI indices over Upper Jhelum Basin computed from the raw (first box in each panel) and bias corrected data (rest of boxes; CW: component-wise, D: direct). Each box represents the interquartile range of biases across all models, which are depicted individually with colored dots (CMIP6 in red, CORDEX in green, CORDEX-CORE in blue), whiskers expand the full range of biases. Red horizontal lines depict perfect performance, for reference.



### 3.5 Effect of bias correction on the spatial pattern of SPEI characteristics

The ability of the BC methods under both approaches to represent the spatial structure of SPEI indices (median duration and severity and absolute frequency of dry and wet events) in the historical period (1986-2005) for the UJB is explored by Taylor diagrams (Figure 6). These Taylor diagrams (Taylor, 2001) show the degree of agreement between the spatial pattern of raw and bias-corrected data and the observed counterpart for SPEI indices, by means of spatial Pearson correlation coefficient (dotted lines), (centered) root mean squared error (blue curves) and normalized standard deviation (black curve denotes perfect performance).

The Taylor diagrams indicate that overall, all BC methods improve upon the raw model output for all datasets and SPEI indices. The correlation coefficient is much lower for duration and severity SPEI indices (typically lies between 0.1 to 0.8) whereas it amounts to between 0.5 and 0.9 for WF and over 0.8 for DF. Concerning the normalized standard deviation (nSD) most of the bias-corrected results underestimate the spatial variability in all SPEI indices. This underestimation amounts up to 50% (nSD between 0.5 to 1.0) except for WD and DF, which show maximum and minimum underestimation, respectively. The centered root mean square errors between BC and reference SPEI indices are found to be in the range of 0.3 to 1.2, being the lowest for frequency indices especially for DF (0.3 to 0.6). Overall, the spatial pattern for the frequency SPEI indices indicates better agreement with the reference dataset than for the duration and severity SPEI indices for most of the BC methods and datasets.

The performance of BC methods is rather consistent for most of the SPEI indices regardless of the statistical measure (i.e., correlation coefficient, normalized standard deviation and centered root mean square error). More specifically, all BC methods under direct approach show better agreement than component-wise approach for most of the SPEI indices and datasets. The EQM, PQM and QDM under the direct approach grouped together for all SPEI indices and datasets. For the component-wise approach, no systematic difference between the best-performing univariate methods and the multivariate ones is found.

Regarding the spatial resolution of the original model data, no clear benefit of the higher resolution (CORDEX-CORE vs. CORDEX and CMIP6) was found, results vary with the SPEI indices and depend more on the BC method than on the model ensemble. The degree of agreement of CORDEX and CORDEX-CORE experiments with reference data is comparable and they tend to group together for the frequency indices, however, differences exist with CMIP6 models. For the best performing BC methods, the CORDEX ensemble presents the largest correlation and smallest root mean square error for dry events and WD, whereas the CORDEX-CORE ensemble represents better the spatial variability. For the dry SPEI indices, CMIP6 falls behind both CORDEX experiments, even for the best performing BC methods. For WS, most datasets present correlation coefficients of 0.4-0.6 and the spatial variability is larger for CORDEX-CORE and CMIP6. For WF, datasets group by BC method, regardless of the model ensemble. Note that here the multi-model ensemble mean is considered, which might hinder the potential added value of individual simulations.

To further explore the inner-ensemble variability and potential added value of individual simulations of CORDEX and CORDEX-CORE experiments, Fig. 7 shows the performance of bias-corrected (using D-EQM) individual climate model simulations from the two experiments. Interestingly, there is no clear best performing driving GCM or RCM for all SPEI indices and large discrepancies with the reference data remain for some individual simulations after bias correction. In general, CORDEX simulations show higher correlation coefficient and smaller root mean squared error and CORDEX-CORE presents more accurate spatial variability. The RCM and GCM combination also matters. For example, the performance of REMO2015 is poor when driven by MPI (which is its typical driving GCM) but it is one of the best with HadGEM2 for most of the indices. On the other hand, HadGEM2-driven RCM under CORDEX-CORE experiment (REMO2015) performed well as compared to CORDEX experiment (RCA4). Similarly, the added value of higher spatial resolution (CORDEX-CORE over CORDEX) can also be seen with RegCM4 driven by MPI under both experiments (dark brown filled and non-filled circle). However, the added value of CORDEX-CORE experiment does not hold for all simulation pairs. For instance, NorESM1-driven RCMs under both experiments do not show a clear behavior (filled and non-filled dark green shapes).



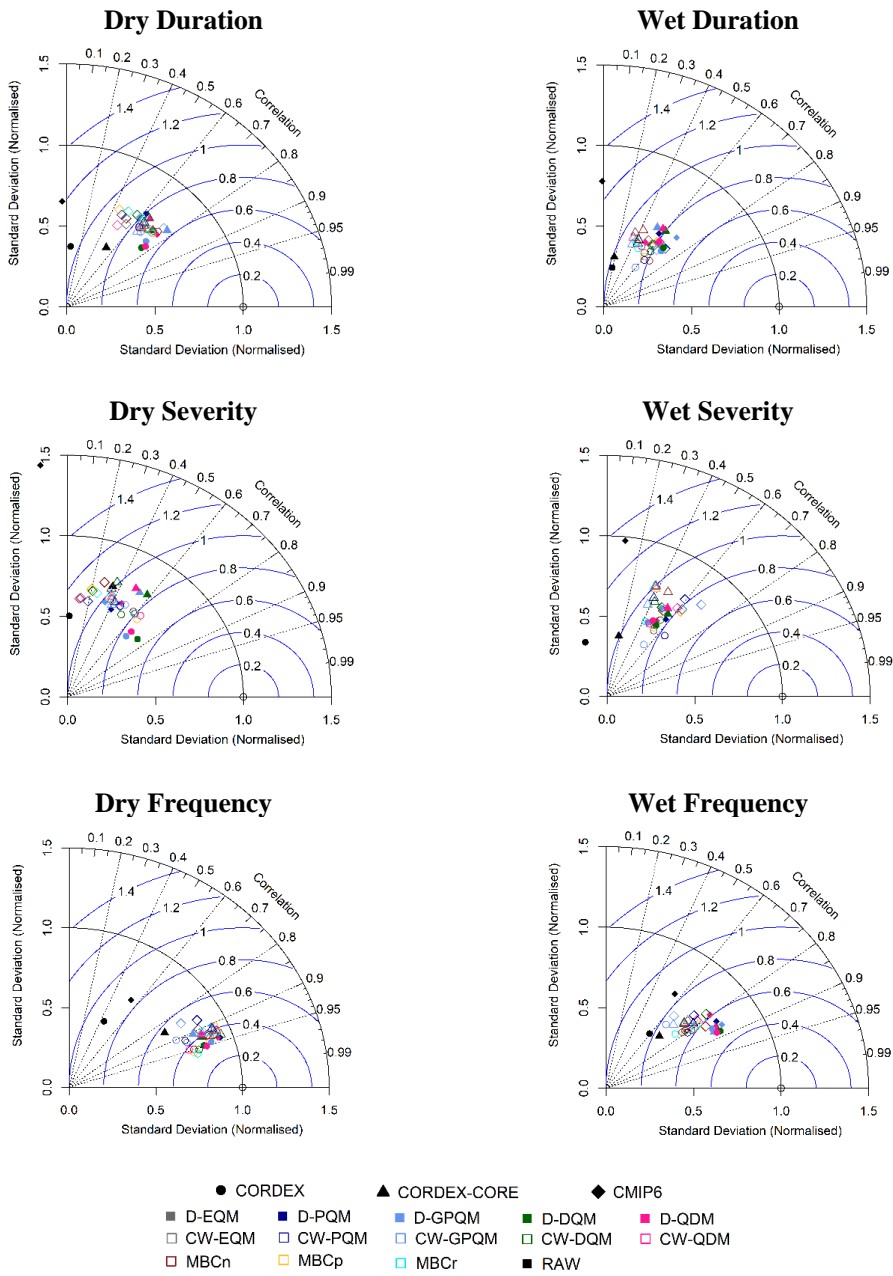

**Figure 6:** Taylor diagrams showing the performance of BC methods (colors) and datasets (markers) with respect to the spatial variability of the SPEI indices in the historical period (1986-2005) for the UJB. Each marker represents the evaluation measures for the multi-model ensemble mean SPEI indices, for the three modelling experiments (CORDEX in circles, CORDEX-CORE in triangles, CMIP6 in diamonds). Filled markers: Direct BC approach, non-filled markers: component-wise BC approach, black marker: raw model output. Note: In case of CMIP6, D-EQM (filled grey diamond) and D-PQM (filled dark blue diamond) are grouped together for all SPEI indices. Note: In case of



CMIP6, D-QDM (filled pink diamond) and D-DQM (filled dark green diamond) are grouped together for all SPEI indices. Note: In case of CORDEX and CORDEX-CORE experiments, D-EQM (filled grey marker), D-PQM (filled dark-blue marker) and D-QDM (filled pink marker) are grouped together for all SPEI indices.



**Figure 7:** Taylor diagrams showing the performance of individual simulations of CORDEX and CORDEX-CORE experiments bias-corrected using D-EQM method, with respect to the spatial pattern of SPEI indices in the historical





period (1986-2005) for the UJB. Filled markers: CORDEX models, non-filled markers: CORDEX-CORE models, 3 different markers show the 3 RCMs under both experiments (i.e., RegCM4, RCA4, REMO2009 from CORDEX and COSMO, REMO2015, RegCM4-7 from CORDEX-CORE); colors show the driving GCMs (same color means same driving GCM); plus marker and cross in black shows the multi-model ensemble mean (MMEm)

for CORDEX and CORDEX-CORE experiment, respectively. **Note:** RCA4 driven by CSIRO (filled blue triangle) and ICHEC-EC-EARTH (filled maroon triangle) are grouped together for all SPEI indices. **Note:** RCA4 driven by MIROC (filled yellow triangle), RCA4 driven by MOHC-HadGEM2 (filled grey triangle), RCA4 driven by MPI (filled cyan triangle) and REMO2009 driven by MPI (filled cyan square) are grouped together for all SPEI indices. **Note:** RegCM4 driven by MPI (filled brown circle) and RCA4 driven by NCC-NorESM1 (filled dark green triangle)

are grouped together.

## 4.0 Discussion and Conclusion

This study assesses the performance of two BC approaches (direct and component-wise) and eight methods (univariate and multivariate) for an impact relevant, multivariate drought index (SPEI) that characterizes wet and dry

extreme events over the Upper Jhelum Basin, in the Western Himalaya.

From obtained results, most of the univariate BC methods under both direct- and component-wise approaches exhibit a comparable performance in current climate, with a slightly better performance for the direct approach. The direct approach with univariate methods provides also comparable results to more sophisticated multivariate methods, which could be due to the weak relationship among the input variables of SPEI in this region. The spatial pattern is

better reproduced by the direct approach than component-wise approach (see Taylor diagrams in Sec. 3.5). This is expected from the experimental design, since under the direct approach, the SPEI is corrected as a single variable, regardless of the biases in the input essential climatic variables and in their interdependencies. Concerning univariate methods, the performance of parametric methods (i.e., PQM and GPQM) is found to be the worst especially in terms of inter-variable physical dependencies. Among the univariate empirical BC methods, both QDM and DQM exhibit a

similar performance for the SPEI indices and inter-variable physical dependencies. Regarding the performance multivariate BC methods, all methods show comparable performance. Overall, the best performing univariate (i.e., empirical methods) are comparable to the multivariate ones.

Findings on the limited performance of few univariate BC methods, especially parametric (PQM and GPQM) under both approaches are admittedly based on inter-variable physical relationship. The direct correction of SPEI

using univariate BC methods, that does not consider the dependencies between variables, has the advantage of adjusting a single variable instead of several variables with different statistical properties, and in this work shows slightly better performance than correcting individual climatic variable prior to the SPEI calculation. Nevertheless, the advantages of direct approach over component-wise, multivariate over univariate BC methods, and trend-preserving methods over non-trend-preserving methods still need to be evaluated for the projected future conditions

(Casanueva et al., 2018). Moreover, the findings of our study may differ for other multivariate hazards or impacts related indices. Since the individual variables in a multivariate hazard index are interlinked differently, the contribution of their individual biases to the associated multivariate hazard index may lead to different results. For instance, biases in wet-bulb globe temperature (WBGT) are found to be smaller than Chandler Burning Index (CBI) for a given model output, yet both indices are based on temperature and relative humidity (Villalobos-Herrera et al., 2021). This is

attributed to the construction of the index since bias in CBI is mainly driven by the bias in relative humidity whereas bias in WBGT interplays between biases in temperature and relative humidity which compensate each other.

Contrasting conclusions are found in the literature about the added value of multivariate BC methods over univariate ones in impact relevant studies. For example, Guo et al. (2020) reported the regionally dependent added value of MBC methods over univariate methods in reproducing observed inter-variable dependencies and observed

streamflow using GR4J hydrological model. Similar findings also have been identified with the Canadian Fire Weather Index (Cannon, 2018), heat-stress (WBGT ) and fire risk (CBI) indicators (Zscheischler et al., 2019), and snowmelt-





driven streamflow (Meyer et al., 2019). On the other hand, Eum et al. (2020) reported marginal improvement of MBC methods in reproducing the extreme climatic indices and hydrological indicators over Alberta, Canada. Räty et al. (2018) also indicated that it is difficult to demonstrate that multivariate methods may significantly reduce biases in hydrologic indicators. Van De Velde et al. (2020) stated that the simpler univariate BC methods are better to use for climate change impact assessment as the MBC methods failed to handle non-stationary climate conditions. In comparison with the direct BC approach, Chen et al. (2021) found similar performance of MBC and the direct correction of hydrological model output i.e., streamflow in present climate, however both are sensitive to non-stationary biases during the validation period. They recommended the MBC especially for streamflow projections under strong anthropogenic signature on the climate. They also reported that biases in streamflow simulations depend on the climate model output, hydrological model, streamflow metrics and region. Similar conclusions were also made by (Casanueva et al., 2018), who state that the direct correction of the Canadian Fire Weather Index (FWI) presents similar performance as the FWI calculated from individually corrected climate variable in present climate but found a more robust behavior for the component-wise approach under future climate change.

The added value of higher climate model resolution (CORDEX-CORE vs. CORDEX and CMIP6) is not evident in the evaluation step, either in terms of inter-variable physical coherence or SPEI indices. There is some indication of added value of CORDEX-CORE with respect to CORDEX and CMIP6 in terms of the representation of spatial variability. However, the CORDEX ensemble performs best in terms of correlation and root mean squared error of the spatial patterns. Nevertheless, the absence of obvious benefits of a finer grid resolution in the present work does not rule out such an added value in general. For instance, a study conducted by Maharana et al. (2021) on Indian summer monsoon precipitation found improved representation of mean precipitation climatology of individual CORDEX-CORE models from the CORDEX experiment. Moreover, the 0.5º resolution of the gridded observational reference (W5E5), coarser than that of the 0.22º CORDEX-CORE simulations, allows us to draw conclusions concerning a lack of large-scale bias improvements by the 0.22º CORDEX-CORE experiments, but hinders the identification of benefits at a smaller scale. The added value of finer spatial resolution could be more obvious if the evaluation is carried out at their original resolution (Prein et al., 2016; Casanueva et al., 2019) especially for the processes over complex terrains where abrupt orographic changes cause much larger spatial variability. Although the uncertainty due to the spatial resolution of the reference dataset is out of scope of the present study due to limited availability of reliable finer resolution observational datasets for the studied region, many other studies acknowledge its greater impact especially for extreme precipitation indices (Casanueva et al., 2019; Casanueva et al., 2020). Kotlarski et al. (2014) also stated that the obvious added value of finer resolution simulations over its coarser counterpart is strongly dependent on the availability and accessibility of fine-gridded and high-quality observational data sets.

To summarize, there is some added value of multivariate bias correction with respect to univariate BC methods in the representation of the inter-variable structures but comparable results to the best performing univariate BC methods are found in terms of biases in SPEI indices. Present climate evaluation shows limited added value of higher spatial resolution simulations, mainly due to the experimental design (both resolutions are remapped onto the 50×50 km observational grid). Furthermore, future work should explore to what extent current results of bias correction are robust under projected climate.





**Code availability:** All calculations and plots were produced using R (version 3.3.2) and ArcMap (version 10.8) by making use of open-source R packages. For univariate bias correction, the present study uses the implementation included in the R package "downscaleR" (version 3.3.3) which is part of the R bundle "climate4R" (Bedia et al., 2020; Iturbide et al., 2019) available from a GitHub repository (https://github.com/SantanderMetGroup/downscaleR, https://zenodo.org/record/5070432). R package "MBC" version 0.10-5 (https://rdocumentation.org/packages/MBC/versions/0.10-5) is used for the multivariate bias correction methods (Cannon, 2020). The potential evapotranspiration (PET) and standardized precipitation evapotranspiration index (SPEI) were calculated with the R package "SPEI" (version 1.7). All the code to perform the derived analyses, calculations and plots are also based on R scripts and ArcMap which are available in (10.5281/zenodo.7296744).

**Data availability:** The reference data (W5E5) are available for download at https://data.isimip.org/10.5880/PIK.2019.023 and the climate model simulations from three initiatives (CORDEX, CORDEX-CORE and CMIP6) used in this study are accessible via the Earth System Grid Federation (ESGF archive; https://esgf.llnl.gov). All datasets were accessed using the R package "loadeR" which is part of the R bundle "climate4R" (Iturbide et al., 2019), available from a GitHub repository (https://github.com/SantanderMetGroup/climate4R). The package is built on the NetCDF-Java API and allows user-friendly data access either from local or remote locations (e.g. OPeNDAP servers) and it is fully integrated with the User Data Gateway (UDG), a Climate Data Service deployed and maintained by the Santander Meteorology Group.

**Author contributions:** R.A., A.C. and G.G. conceptualized the study. R.A. performed the formal analyses. A.C. gave support regarding the bias correction methods and model data and M.U.L. provided ideas for new analyses and illustrations. G.G. and A.C. supervised the work. R.A. wrote the first draft of the paper and all authors reviewed the text and contributed to the final version.

**Competing interests:** The authors declare that they have no conflict of interest.

**Acknowledgment:** The authors acknowledge the World Climate Research Programme's Working Group on Regional Climate, and the Working Group on Coupled Modelling, former coordinating body of CORDEX and responsible panel for CMIP5. We also thank the climate modelling groups (listed in Tables 1 and 2 of this paper) for producing and making available their model output. We also acknowledge the Earth System Grid Federation infrastructure an international effort led by the U.S. Department of Energy's Program for Climate Model Diagnosis and Intercomparison, the European Network for Earth System Modelling and other partners in the Global Organisation for Earth System Science Portals (GO-ESSP). Data were accessed through the Santander Climate Data Service, which is maintained by the Santander Meteorology Group. The authors are also grateful to Ezequiel Cimadevilla (Santander Meteorology Group) for technical support.

**Financial support:** The first author as a PhD student received funding from the Cooperation Agreement PFK PhD program 2019–2022 "Partnership for Knowledge-Platform 2: Health and WASH (WAter Sanitation and good Hygiene)" of the AICS-Italian Agency for Development Cooperation to attend higher education programs in Italy in favor of non-Italian citizens. The first author also thanks the Erasmus Trainingship Program which allowed a research stay of 3 months at the University of Cantabria, Spain. A.C. acknowledges support from Project COMPOUND Grant TED2021-131334A-I00 funded by MCIN/AEI/ 10.13039/501100011033 and by the European Union NextGenerationEU/PRTR and the Horizon 2020 Project IS-ENES3 (grant agreement No 824084).





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
