# Peer review of "Evaluation of bias correction methods for a multivariate drought index: casestudy of the Upper Jhelum Basin"

_Geoscientific Model Development, 2022_

## Author Comment (AC2)

**Evaluation of bias correction methods for a multivariate drought index: case-study of the Upper Jhelum Basin**

**Response to Referee 1**

**General Comments:**

Dear Authors,

This is a very interesting paper on the topic of assessing bias-adjustment techniques, where I am also doing work in. It is relevant to the scope of the journal and could be a useful resource for the community for comparing the performance of univariate bias-adjustment methods vs. multivariate methods in the context of multivariate climate indices. However, there are a number of points that should be further discussed before publication is recommended:

**Response:** We thank the referee for the time devoted to review our manuscript, and the positive feedback provided. Along the next lines, the different comments posed by the reviewer are reviewed point by point. The referee's comments are indicated in black, and the author responses in red fonts.

**Comment:** Please provide a comment or two to describe to readers as to why SPEI and the Upper Jhelum Basin are appropriate for testing univariate and multivariate bias-adjustment approaches.

**Response:** Thank you for your comment.
The proposed framework is demonstrated as a case study over a transboundary watershed, namely the Upper Jhelum Basin (UJB) located at the foothills of Western Himalaya, one of the most affected mountainous ranges by climate change. The region has already witnessed an increase in extreme hydro-meteorological events in the last few decades (Pachauri et al., 2014), and hence the projection of these extreme events cannot be left apart in the development of the climate change adaptation strategy for the region. The use of SPEI over other drought indices such as SPI is preferred due to its link to potential evapotranspiration (PET), which makes it more sensitive in the context of global warming (Vicente-Serrano et al., 2010; Huang et al., 2017; Yao et al., 2018).

The aforementioned information about the motivation of the study over UJB is added in the Introduction section (page 3, lines 20:27).

**Comment:** Based on the results of the bias analysis and the Taylor diagrams, even though biases are reduced via the various methods it does not appear that SPEI is resolved well over all. Is the 50km resolution sufficient in resolving the topography of the region?

**Response:**
We checked that BC methods worked as expected and improved upon raw climate model outputs considering several statistics, for instance in terms of similarity of daily distributions (Perkins skill score, Fig. 1) and in terms of biases in temporal indices, which have not been calibrated by any of the BC methods (Figs. S1-S5 in the revised Supplementary Material). For the latter we analyzed univariate temporal indices from the EU-COST Action VALUE since they could shed light on SPEI biases, (namely transition

probability of a wet day given that the previous day was dry, longest dry spell, longest warm spell, amplitude of seasonality and interannual variance; (Maraun et al., 2019). We found an overall reduction in biases of these indices for all BC methods except GPQM and DQM (especially for precipitation), which present the largest departures from the reference dataset. Although remaining biases in interannual variance of temperatures are relatively high for temperature for few months, the overall performance is considered to be reasonable.

We agree that the improvement in the SPEI and derived indices is not evident after BC. We argue that this is mainly due to remaining biases in univariate properties (such as the above-mentioned temporal ones) and to high non-linearities in the multivariate index calculation (note that SPEI and derived metrics are not directly calibrated by any of the BC methods), but some benefits of BC are found for biases in the frequency of dry and wet extremes and in the representation of spatial patterns.

We added a little discussion about it in the revised manuscript. (See highlighted text in Sect. 3.1 (page 9, lines 35:48), section 3.2 (page 10, lines 12:18) and Discussion section (page 21, lines 27:31 and page 22, lines 10:17)).

The spatial resolution at which the whole analysis has been carried out is limited by the availability of observational data at high spatial resolution, which is the cornerstone of BC. Moreover, we aimed at using the largest available climate model ensemble, which is nowadays provided by CORDEX (at approximately 50km resolution) and CMIP (varying resolution depending on the model). Note that a smaller set of simulations derived at 25km resolution is provided by CORDEX-CORE which was here upscaled to the 50km CORDEX grid since no observational dataset was available for BC, but we assume that some indication of the potential added value of the high resolution could be present at the coarse scale (see highlighted text in Section 2.6 Experimental Framework (page 8, lines 40:47)).

**Comment:** Please justify more explicitly/clearly that these findings (that both univariate and multivariate methods for SPEI perform similarly/well) is applicable to different regions in the world, input variables, and/or multivariate indices.

**Response:** The applicability of our results to different regions and multivariate indices is explained with example in the discussion section (page 21, lines 39:46).

**Specific Comments:**

**Comment:** Page 4, line 5: I was not familiar with SPEI, so it was not immediately apparent how the Ra parameter was derived in the simplified Hargreaves-Samani equation. It would be useful for readers who aren't familiar with SPEI to indicate that the radiation parameter is derived using the latitude of the site/grid.

**Response:** Thank you for your comment. The aforementioned information is added in the revised manuscript (see highlighted text in the section 2.1 Standardized Precipitation Evapotranspiration Index-SPEI (page 4, lines 11:12)).

**Comment:** Page 4, line 16: Extremes events are defined as SPEI values $\geq +1$ and $\leq -1$ in this paper. Please comment on why these values were chosen – were there any past studies that also defined extreme events using these thresholds?

**Response:** The thresholds for extreme events were taken from the literature (Svoboda et al., 2012). We cited this work in the revised version of the manuscript (see highlighted text in the section 2.2 Identification of extreme events and their characteristics, (page 4, lines 25)).

**Comment:** Page 4, line 31: Why were only 20 years (1986-2005) used as the historical and not a slightly longer 30 year period? I believe W5E5v1.0 was available from 1979-2016.

**Response:** Yes, W5E5v1.0 is available from 1979-2016. However, our motivation towards the selection of 20 years calibration period is to maximize the number of climate model simulations for the present study. Moreover, this 20-years period is typically used as the reference historical period in many studies, in particular in the IPCC fifth Assessment Report for the calculation of climate change signals and our final objective is to assess changes in future climate compared to the historical simulations in a subsequent study. We included this information in the revised manuscript (see highlighted text in the section 2.3 Reference Dataset (page 4, lines 40:42)).

**Comment:** Page 8, section 2.6: When applying these methods, did you aggregate/pool the daily data into month-of-year/seasonal windows before bias-adjustment to account for precipitation biases in the seasonal cycle? Likewise, for MBCn, how many iterations were used? Could you describe in more depth how you applied these quantile mapping algorithms?

**Response:** We are sorry for the confusion. All BC methods are calibrated in the period 1986–2005 using daily time series, being the correction functions calculated separately for each month separately in order to account for biases varying along the year. We clarified this information in the section "Experimental framework".
Secondly, the present study shows the results of MBCn following 30 iterations. We added this information in the revised section "Multivariate Bias Correction Methods".

**Comment:** Page 8, line 46-48: Could you justify why nearest neighbor interpolation was chosen over other methods such as bilinear/cubic? Can you verify whether the "added value of the higher resolution WAS-22" is still present after remapping to the coarse resolution?

**Response:** The highest resolution mismatch is found between W5E5 (50km) and CORDEX-CORE (25km), therefore CORDEX-CORE simulations were conservatively remapped into the observational grid in order to guarantee the representation of areal averages. Thus, any potential added value of the high resolution could transfer to the upscaled data, i.e., details related to better resolved local processes from high resolution cannot be discerned but may be still present after smoothing them onto the coarse resolution. Thus, we addressed the added value of the high resolution at its skillful scale (Grasso, 2000), which is known to be coarser than the scale in which the simulation was develop. This is the aspect we try to verify when comparing results for CORDEX and CORDEX-CORE. For CMIP6 and CORDEX simulations, the nearest neighbor gridbox to each observational gridbox was taken, since the resolution mismatch is small. This interpolation method maintains the higher spatial variability of the topographical areas whereas bilinear or cubic interpolations would smooth the spatial patterns.
We clarified the different procedures in the revised section "Experimental framework" (page 8, lines 40:47).

**Comment:** Page 9, line 1: Were the issues of temperature reversals (i.e., Tmin>Tmax) considered, and/or how did you resolve this? Based on Thrasher et al. (2012), temperature reversals may be encountered post

bias-adjustment, while Cannon et al. (2021) multivariate bias-adjusted the daily diurnal cycle and Tmean before deriving Tmax & Tmin to ensure reversals are avoided.

**Response:** Yes, we checked the issue of temperature reversals (i.e., Tmin>Tmax) after bias correction and found that a negligible fraction of days had this issue (maximum of 0.5-1.4% depending upon climate model and BC method, for seldom grid boxes).

**Comment:** Page 13, line 2-3: Is there a reason why the mean biases are expressed as a ratio and not as a delta?

**Response:** There is no specific reason for the representation of mean biases as ratio. We followed the articles from EU-COST Action VALUE series in which different downscaling methods have been evaluated considering several aspects of extreme indices (e.g. (Maraun et al., 2017; Gutiérrez et al., 2019). Further, the representation of biases as ratio makes it easier to know the under- or over-estimation at the first glance.

**Comment:** Page 13, line 22: It is unclear what "partially elevated" means in this context, please clarify.

**Response:** Sentence has been rephrased to make it clearer.

**Comment:** Page 15, line 7: "[…] but still shorter events than in the reference dataset are found after the corrections." Awkward way of phrasing, or possibly a strange placement for the word "still".

**Response:** Correction has been carried out.

**Comment:** Page 18-20, Figures 6 & 7: The study area spans over 30 grid cells at a 50km resolution – are these large enough of a sample size to use for spatial analysis via Taylor diagrams?
**Response**: This is certainly a good point. We acknowledge that 30 grid boxes might be a bit too few for statistical analyses but still think that these plots are a nice summary for multi-model ensembles like in the present study. From a statistical point of view, the sample size required to determine whether a correlation coefficient of 0.5 differs from zero is of 29, using 0.2 as probability of type II error (beta) and a threshold probability for rejecting the null hypothesis (alpha) of 0.05 (Bujang and Baharum, 2016). Thus, correlations below 0.5 might not be statistically significant given the small sample size. This issue is mentioned in the revised Section 3.4 (page 17, lines 7:8).

**Technical corrections:**
Page 3, line 30+31: Section numbering should be 2 and 2.1
Page9, line 12: Heading should be spaced after text
Page 20: Legend for model names is a bit fuzzy when zoomed in – would it be possible to have this at a higher resolution?
Figures 3, 4, S1, S2, S3: Lower (left) bound value of the color bar is not equal in increment to the others.
Throughout the paper (e.g., Page 4 line 11; Page 22 line 9): citation formatting, i.e., brackets should just be around the publication year, like other examples throughout the manuscript when authors are directly addressed in the sentence.

**Response:** Thanks for the comments. All corrections have been carried out. We are sorry about the figure's low resolution, we have improved them and will submit them as separate files.

**References**

Bujang, M. A. and Baharum, N.: Sample size guideline for correlation analysis, World, 3, 37-46, 2016.

Grasso, L. D.: The differentiation between grid spacing and resolution and their application to numerical modeling, Bulletin of the American Meteorological Society, 81, 579-580, 2000.

Gutiérrez, J. M., Maraun, D., Widmann, M., Huth, R., Hertig, E., Benestad, R., Rössler, O., Wibig, J., Wilcke, R., and Kotlarski, S.: An intercomparison of a large ensemble of statistical downscaling methods over Europe: Results from the VALUE perfect predictor cross-validation experiment, International journal of climatology, 39, 3750-3785, 2019.

Huang, C., Zhang, Q., Singh, V. P., Gu, X., and Shi, P.: Spatio-temporal variation of dryness/wetness across the Pearl River basin, China, and relation to climate indices, International Journal of Climatology, 37, 318-332, 2017.

Maraun, D., Widmann, M., and Gutiérrez, J. M.: Statistical downscaling skill under present climate conditions: A synthesis of the VALUE perfect predictor experiment, International Journal of Climatology, 39, 3692-3703, 2019.

Maraun, D., Shepherd, T. G., Widmann, M., Zappa, G., Walton, D., Gutiérrez, J. M., Hagemann, S., Richter, I., Soares, P. M., and Hall, A.: Towards process-informed bias correction of climate change simulations, Nature Climate Change, 7, 764-773, 2017.

Pachauri, R. K., Allen, M. R., Barros, V. R., Broome, J., Cramer, W., Christ, R., Church, J. A., Clarke, L., Dahe, Q., and Dasgupta, P.: Climate change 2014: synthesis report. Contribution of Working Groups I, II and III to the fifth assessment report of the Intergovernmental Panel on Climate Change, Ipcc2014.

Svoboda, M., Hayes, M., and Wood, D.: Standardized precipitation index: user guide, 2012.

Vicente-Serrano, S. M., Beguería, S., and López-Moreno, J. I.: A multiscalar drought index sensitive to global warming: the standardized precipitation evapotranspiration index, Journal of climate, 23, 1696-1718, 2010.

Yao, J., Zhao, Y., Chen, Y., Yu, X., and Zhang, R.: Multi-scale assessments of droughts: a case study in Xinjiang, China, Science of the Total Environment, 630, 444-452, 2018.

---

## Author Comment (AC3)

**Evaluation of bias correction methods for a multivariate drought index: case-study of the Upper Jhelum Basin**

**Response to Referee 2**

In their study, the authors of "Climate change projections of wet and dry extreme events in the Upper Jhelum Basin using a multivariate drought index: Evaluation of bias correction" assess the performances of several univariate (2) and multivariate (8) bias correction methods applied to climate models outputs for impact studies. The multivariate drought index SPEI is considered to evaluate the adjustments of wet and dry extreme events over the Upper Jhelum Basin, in the Western Himalaya region. Two experiments of bias correction are performed in this study: 1) the component-wise approach that consists in applying bias correction (BC) methods prior to the computation of the SPEI index, and 2) the direct approach that consists in calculating the multivariate SPEI index first, and then adjusting it using univariate BC methods. Corrections are performed to adjust daily maximum temperature, minimum temperature and precipitation from several CMIP6 GCMs, CORDEX and CORDEX-CORE RCMs with different spatial resolutions over the historical period (1986-2005). Corrections are evaluated in terms of inter-variable relationships and SPEI characteristics on the same historical period with respect to W5E5 reanalysis dataset. The authors find that the multivariate BC methods have some added value over univariate BC methods concerning the adjustment of inter-variable properties. However, both univariate and multivariate methods present similar performances for the correction of SPEI indices. The direct approach shows slightly better results than the component approach and no added value was obtained when considering high resolution products.

The article is scientifically interesting, its structure is clear and easy to follow, the results are well explained and summarized. I think this study certainly falls within the scope of the journal. However, I think there are a few minor issues that should be considered to improve the study before publication.

**Response:** We thank the referee for the time devoted to review our manuscript, and the positive feedback provided. Along the next lines, the different comments posed by the reviewer are reviewed point by point. The referee's comments are indicated in black, and the author responses in red fonts.

**General comments:**

**Comment:** SPEI has been chosen to evaluate univariate and multivariate bias correction methods. In this study, the index is computed in several steps involving precipitation, Tmax and Tmin time series at different time scales. Thus, not only a good representation of the values (marginal properties) and dependence (inter-variable properties) of the variables is important for the SPEI index, but also the temporal properties. None of the multivariate BC methods used in this study are designed to adjust temporal properties. Moreover, multivariate BC methods can also deteriorate temporal properties (e.g., François et al., 2020). Consequently, it is not clear in this study whether the comparable performances of multivariate BCs with respect to univariate BCs are due to compensating effects between improvement of inter-variable properties with multivariate BC and deterioration of temporal properties at the same time. Would the same conclusions be obtained by considering other multivariate indices than SPEI, e.g., indices for which inter-variable properties are important, but not temporal ones? I think that these points (1. importance of temporal properties for SPEI, 2. inability of the implemented BC methods to adjust temporal properties and 3.

potential deterioration of temporal properties by multivariate BCs) should at least be mentioned in the discussions to provide some nuances to the conclusions of this study as explained above.

**Response:** Thank you for your comment. Temporal aspects have a role in the SPEI derived indices (since SPEI was calculated using a 30-day accumulation period at daily time step and extreme events were defined considering consecutive SPEI monthly values) and thus they might help to explain biases in SPEI indices. To check the ability of BC methods to adjust the temporal properties of the corrected time series, we evaluated all BC methods considering several univariate indices from the EU-COST Action VALUE which are specifically related to temporal properties. More precisely, we considered the transition probability of a wet day given that the previous day was dry, the longest dry spell, the longest warm spell, the amplitude of the annual cycle and interannual variance (Maraun et al., 2019), which have not been calibrated by any of the BC methods. We found a large reduction in biases of these indices with all BC methods except for GPQM and DQM (especially for precipitation) and no clear benefit of multivariate methods (see Figures S1-S5 in the revised Supplementary Material and new Section 3.1 in the revised manuscript). Although biases in the interannual variance of temperatures are relatively high for few months, the overall performance is considered to be reasonable.
In addition to this new evaluation of univariate temporal properties, we also looked at the correlation between pairs or variables at monthly timescale and found high correlations between precipitation and temperature in the reference dataset in some parts of the domain (up to -0.54, see Fig. S7 in the revised Supplementary Material), despite the low correlations at daily scale (Fig. S1). These correlations are not present in most of the raw models but are improved after BC (except for GPQM, see Fig. S8), with no evident added value of multivariate methods. Still, we focus on daily statistics (correlation and PSS) in the main manuscript since this is when inter-variable relationships are expected to be relevant for the SPEI calculated in the present work.
We incorporated these new analyses in the revised manuscript (See highlighted text in Sect. 3.1 (page 9, lines 35:48), section 3.2 (page 10, lines 12:18) and Discussion section (page 21, lines 27:31 and page 22, lines 10:17)).

**Comment:** I really like the title which is clear, but I find the term "climate change projections" a bit misleading. The historical period 1986-2005 is only considered in the study, and thus authors are not looking into climate projections, i.e., simulations of future evolutions of the climate system. The notion of "climate change" is also misleading as the changes of the climate system are not particularly investigated in this study, not even those that could have occurred during the 1986-2005 period. I would propose to find another title for the study avoiding the words "climate change projections".

**Response:** Thanks for the comment. The reason for the name is that this work corresponds to the first part of a wider study where future climate change projections are analyzed. However, we understand that it can be misleading as it is, thus the title has been changed to "Evaluation of bias correction methods for a multivariate drought index: case-study of the Upper Jhelum Basin" in the revised manuscript.

**Comment:** In Section 4 - Discussion and conclusion: I really like the way results are summarized and discussed. However, I think it would be interesting to detail a bit more about future research by adding a few sentences. Besides investigating the robustness of results under climate change as already mentioned at the very end of the study, what are the next steps of this work? I think it would be helpful to mention a few avenues of research, as it would help to better connect this work to the research community.

**Response:** As for the future work, we had several lines at the time of submission, so we preferred to keep it general. However, we have incorporated your suggestion into the revised manuscript since the main draft of the future work has been drawn.
We incorporated this information in the revised manuscript (Discussion section, page 23, lines 1:6).

**Specific comments:**

- Page 2, L21: "The use of raw GCM and RCM output for subsequent impact studies without any post processing could lead to ill-informed adaptation decisions for the foreseeable future.". Do you have any examples/references to support this sentence?
- Page 4, L20: "above/below 1". I assume you meant "above 1 and below -1"?
- Page5, Table 1 and 2, and Section 2.4: I don't understand why you detail the scenarios for the different models. As your study is focused on the 1986-2005 period, it seems that there is no particular reason to provide such details. It might be preferable to remove some parts of the text and tables mentioning information on scenarios.
- Page 5, Table 1: It might be preferable to round resolution numbers in the table.

**Response:** Many thanks for the comments. The suggested corrections have been carried out. Regarding the first comment, a few references have been added (Piani et al., 2010; Haerter et al., 2011; Argüeso et al., 2013) and the sentence has been slightly rephrased (page 2, lines 19:21).

**References:**

Argüeso, D., Evans, J., and Fita, L.: Precipitation bias correction of very high resolution regional climate models, Hydrology and Earth System Sciences, 17, 4379-4388, 2013.

Haerter, J., Hagemann, S., Moseley, C., and Piani, C.: Climate model bias correction and the role of timescales, Hydrology and Earth System Sciences, 15, 1065-1079, 2011.

Maraun, D., Widmann, M., and Gutiérrez, J. M.: Statistical downscaling skill under present climate conditions: A synthesis of the VALUE perfect predictor experiment, International Journal of Climatology, 39, 3692-3703, 2019.

Piani, C., Haerter, J., and Coppola, E.: Statistical bias correction for daily precipitation in regional climate models over Europe, Theoretical and applied climatology, 99, 187-192, 2010.

---

## Author Response (AR2)

**Evaluation of bias correction methods for a multivariate drought index: case study of the Upper Jhelum Basin**

**We thank the Topical Editor for taking the time to review our manuscript. Please, find below your comments and our author response, the latter highlighted in red.**

**Comments**
**Public justification (visible to the public if the article is accepted and published**):
Dear author,

Thank you very much for your careful revision of your manuscript which fully address all the referees' remarks.

However, as I already asked you, you have to make improvements to code and data availability aspects before your manuscript can be considered for publication in GMD.

For the code availability, you state that the code to perform the derived analysis, calculations and plots are also based on R scripts and ArcMap which are available from the corresponding author upon request but this is not acceptable for GMD. The code used to produce the results discussed in the paper must be available either from a supplement or from an archive with a digital object identifier (DOI), e.g. ZENODO for model code.

For the Data availability, you mention that the R package "loadeR" used to access the data is available from a GitHub directory but again, this is not acceptable for GMD. All packages used to produce the results discussed in the paper must be available either from a supplement or from an archive with a digital object identifier (DOI), e.g. ZENODO. Please see details at https://www.geoscientific-model-development.net/policies/code_and_data_policy.html.

Thank you for considering these remarks,
With best regards,
Sophie Valcke
**Author Response:** Thank you for your comment. I update the heading "code availability" and "data availability" according to journal requirements. Kindly see the revised version of the manuscript.

Kind regards,
Rubina Ansari